# Recurrence of Idiopathic Membranous Nephropathy in the Kidney Allograft: A Systematic Review

**DOI:** 10.3390/biomedicines12040739

**Published:** 2024-03-26

**Authors:** Anastasios Panagakis, Ioannis Bellos, Konstantinos Grigorakos, Stylianos Panagoutsos, Ploumis Passadakis, Smaragdi Marinaki

**Affiliations:** 1Renal Transplantation Unit, Nephrology Department, Medical School, National and Kapodistrian University of Athens, Laiko Hospital Athens, 11527 Athens, Greece; bellosg@windowslive.com (I.B.); smaragdimarinaki@yahoo.com (S.M.); 2Independent Researcher, 12 Protopappa Avenue, 16345 Athens, Greece; gk_pediatr@yahoo.gr; 3Department of Nephrology, Medical School, Democritus University of Thrace, Dragana Campus, 68100 Alexandroupolis, Greece; spanagou@med.duth.gr (S.P.); ploumis@hol.gr (P.P.)

**Keywords:** idiopathic membranous nephropathy, recurrence, renal allograft, systematic review

## Abstract

Introduction: The recurrence of idiopathic membranous nephropathy (iMN) after kidney transplantation is common, although its exact clinical significance remains unclear. This systematic review aims to elucidate the effects of iMN recurrence on graft survival. Materials and methods: A literature search was performed by systematically searching Medline, Scopus, Web of Science, and Google Scholar from inception. Cohort studies examining iMN recurrence after kidney transplantation were deemed eligible. Meta-analysis was performed by fitting random-effects models. Results: Twelve (12) articles published from 1995 to 2016 reporting on 139 transplant patients with recurrent iMN were included. The median time of the diagnosis of recurrent iMN was 18 months during follow-up from 35 to 120 months. Risk factors for iMN recurrence in the renal allograft are a positive serum test for anti-PLA2R antibodies pretransplant, female sex, younger age, high proteinuria pretransplant, the longest interval from initial disease to end-stage chronic kidney disease, and the combination of alleles HLA DQA1 05:01 and HLA DQB1 02:01. In the pretransplant period, 37 (26.61%) patients had a positive serum test and 18 (12.94%) patients had a positive biopsy stain for anti-PLA2R antibodies. The sensitivity of the pretransplant positive serum test for these antibodies ranges from 57% to 85.30% and the specificity is 85.10–100%. A total of 81.80% of patients who received rituximab as treatment for iMN recurrence achieved complete and partial remission, while 18.20% had no response to treatment. iMN recurrence was not associated with significantly different rates of graft loss (odds ratio = 1.03, 95% CI: 0.52–2.04, *p* = 0.524, I^2^ = 0.00%). Recurrence of iMN was not associated with increased risk of graft loss independently of whether patients were treated with rituximab (OR: 0.98, 95% CI: 0.39–2.50, I^2^: 0%) or not (OR: 1.22, 95% CI: 0.58–2.59, I^2^: 3.8%). Patients with iMN recurrence who achieved remission had significantly reduced risk of graft loss (OR: 0.14, 95% CI: 0.03 to 0.73). Conclusion: The main outcome from this systematic review is that there is no statistically significant difference in graft survival in patients with iMN recurrence compared to those without recurrence in long-term follow-up. The achievement of remission is associated with significantly reduced risk of graft loss.

## 1. Introduction

Membranous nephropathy is the most common cause of nephrotic syndrome in adults and affects all ethnicities, sexes, and ages [1]. Depending on the etiology, it is found in two forms: “primary” or “idiopathic”, where no etiology can be found, and “secondary”, which can be attributed to medications, neoplasms, or other systematic diseases [2]. Idiopathic membranous nephropathy (iMN) is the most common cause of idiopathic nephrotic syndrome in non-diabatic male patients worldwide and contributes to 20–37% of kidney biopsies performed due to proteinuria [3]. Antibodies against the podocyte M-type phospholipase A2 receptor (PLA2R) are present in most cases (>75%) at diagnosis of iMN [4]. At 10-year follow-up of iMN, approximately one third of patients are disease free, one third have partial remission with persistent proteinuria, and one third progress to kidney failure despite immunosuppressive therapy [5]. Recurrence of the primary disease In the graft is the third most common cause of renal allograft loss [6], and like many other glomerulopathies, iMN can recur in the renal allograft. Clearly, this has a negative impact, as it leads to proteinuria [7]. The recurrence rate of iMN is 10–45%, or even higher in centers that perform protocol biopsies on renal grafts [1,7]. Having a high titer of PLA2R antibodies before transplantation is considered a risk factor for iMN recurrence in the kidney allograft, and research is trying to identify other risk factors for recurrence [8]. The introduction of rituximab to the existing immunosuppressive therapy is considered the best intervention. It is the most effective treatment of recurrent iMN, as complete or partial remission of the disease has been achieved in 80% of patients [8].

Current evidence remains inconclusive regarding the exact effects of iMN recurrence after kidney transplantation on long-term outcomes. To this end, the present systematic review and meta-analysis accumulated the existing knowledge in the literature in the field, aiming to assess the influence of iMN recurrence on graft survival, as well as to evaluate the role potential risk factors and treatment strategies.

## 2. Materials and Methods

### 2.1. Study Design

The present systematic review was designed following the PRISMA (Preferred Reporting Items for Systematic Reviews and Meta-Analyses) guidelines [9]. The protocol of the review has been prospectively registered in PROSPERO (ID: CRD42022290026). No ethical approval was required since no new patients were recruited. No amendments to the information from the included studies were performed. The PRISMA checklist is shown in Appendix A.

### 2.2. Eligibility Criteria

The population of the systematic review consisted of adult patients with iMN who underwent kidney transplantation. The main exposure was the recurrence of membranous nephropathy in the kidney allograft and the primary outcome of interest was graft survival. Secondary questions included the time to recurrence of iMN after transplantation, risk factors for recurrence, the role of positive serology for anti-PLA2R antibodies, and the effect of rituximab treatment on graft survival. Both prospective and retrospective cohort studies were deemed eligible. Cross-sectional descriptive studies, small case series, case reports, animal studies, and in vitro studies were excluded.

### 2.3. Literature Search

A literature search was performed by systematically searching the following databases from inception: Medline, Scopus, and Web of Science. Google Scholar, as well as the full reference lists of the retrieved studies (“snowball” method) were also screened to identify potentially missing articles. The date of the last search was set as 31 May 2022. No language restrictions were applied. The search algorithm was based on combinations of the following key-terms: “kidney transplantation”, “renal transplantation”, “allograft”, “membranous nephropathy”, “membranous nephritis”, “recurrence”, “post transplantation”, and “rituximab.” The full search strategy is presented in Appendix A.

### 2.4. Study Selection

The selection of studies followed a 3-stage approach. First, the titles and abstracts of all electronic records were screened to assess for potential eligibility. Subsequently, all articles that were considered to fulfil the inclusion criteria were retrieved as full texts. Then, the studies that met any of the exclusion criteria or did not report any outcome of interest were excluded. The process of study selection was conducted independently by two authors, resolving any disagreements by discussion with a third one.

### 2.5. Data Extraction

The following parameters were extracted from the included studies: study details, clinical and demographic characteristics, recurrent time data, and treatment and outcome data. In more detail, study details included the authors’ names, year of publication, study design, country, and sample size. The clinicodemographic parameters included median age, sex, ABO incompatibility status, type of kidney transplant donor (living or deceased), presence of biopsy indicative of iMN in native kidney before transplantation, date of transplantation, and type of induction therapy and immunosuppressive therapy. Data on iMN recurrence related to whether biopsy was performed for its diagnosis based on protocol or clinical indications, whether PLA2R staining was performed at biopsy, time to disease recurrence since transplantation, serum creatinine, and proteinuria. Data regarding treatment of recurrent iMN included the following: administration of non-immunosuppressive therapy such as angiotensin-converting enzyme inhibitors (ACEi), administration of rituximab, and the regimen. Finally, the following information concerning the outcomes of interest was collected: complete or partial remission, time from treatment to response, graft loss and time to graft loss from initiation of treatment, and relapse and death without disease remission and with a functioning graft. Data extraction was completed in pre-piloted forms by two authors independently, resolving any discrepancies by the consensus of all authors.

### 2.6. Quality Assessment

The risk of bias of the included studies was assessed using the ROBINS-I tool, adjusted for exposure studies [10]. The following domains were taken into account: confounding, classification of exposures, selection of participants, deviations from intended exposures, missing data, measurement of outcomes, and selection of the reported result. Risk of bias evaluation was performed independently by two researchers and any potential discrepancies were resolved through their consensus.

### 2.7. Statistical Analysis

Data analysis was conducted in R-3.6.5 (“metafor” package) [11]. The two-sided *p*-value threshold of 0.05 was used to define statistical significance. The odds ratios (ORs) derived from univariable or multivariable models of the original studies were used in the analysis. Meta-analysis was performed by fitting random-effects models by applying the maximum likelihood method due to the expected inter-study heterogeneity. Statistical heterogeneity was quantified by the inconsistency index (*I*^2^), with values over 50% indicating remarkable heterogeneity [12]. The 95% prediction intervals were calculated to assess the effects to be expected by future studies in the field. The risk of publication bias was evaluated by the visual inspection of funnel plots. Statistical asymmetry tests were not used due to the small number of included studies [13].

## 3. Results

### 3.1. Study Selection

The process of study selection is schematically illustrated in Figure 1. Overall, the literature search resulted in 695 electronic records. After the removal of duplicates, 351 articles were screened for eligibility and 64 of them were retrieved in full-text form. Subsequently, a total of 52 studies were excluded due to the following reasons: case report/series design (*n* = 29), insufficient data about the outcomes of interest (*n* = 9), overlap with other glomerulopathies (*n* = 6), mixed pediatric and adult data (*n* = 5), and mixed recurrent and de novo iMN cases (*n* = 3). As a result, the systematic review was based on 12 studies comprising 139 patients with recurrent iMN.

### 3.2. Quality Assessment

The quality of all included studies was evaluated with the Risk Of Bias In Non-randomized Studies (ROBINS-E) assessment tool, which assesses the potential bias due to confounding, selection, classification, deviation from intended exposures, missing data, measurement, and reporting of the outcomes. Overall, the risk of bias was evaluated to be low in 1 study and moderate in 11 studies. Specifically, there were some concerns due to confounding or missing data, classification, and deviations from intended exposures. The outcomes of risk of bias assessment are presented in Appendix A.

### 3.3. Included Studies

The main methodological characteristics of the included studies are presented in Table 1. Overall, 1 prospective and 11 retrospective cohort studies were included that evaluated 411 patients with a diagnosis of iMN and a total of 144 kidney grafts with iMN recurrence, as some patients received a transplant more than once. The main study location was the USA (6 studies, 91 patients), followed by France (2 studies, 15 patients) and Italy (1 study, 12 patients). The clinical and demographic characteristics of patients with recurrent iMN are summarized in Appendix A.

The majority of patients were males (71.5%) and 72.4% of transplantations were living-donor ones. Transplantation with ABO blood group incompatibility was reported in two patients who received rituximab, and one of them also underwent plasmapheresis. The majority of participants (98.5%) had a biopsy diagnostic of iMN in the native kidney, while in nine cases this information was missing. The diagnosis of iMN recurrence in the kidney allograft was established mainly by biopsy based on clinical indications (proteinuria/worsening kidney function) in 66.4%, while the remaining biopsies were performed based on protocol. At the time of recurrence, the mean serum creatinine was 2.00 mg/dL and the mean urinary protein excretion was 3.97 g/24 h. Importantly, 47.8% of patients were treated for disease recurrence with rituximab. The main administration regimens were four doses of 375 mg/m^2^ per week (8 patients), a single dose of 1 g (2 patients), two doses of 1 g with an interval of 2 weeks (19 patients), and one to two doses of 375 mg/m^2^ with an interval of 2 weeks (6 patients). The studies included in the systematic review showed in 134 grafts that the median time to diagnosis of iMN recurrence in the renal allograft is 18 months during follow-up ranging from 35 to 120 months (Table 2).

### 3.4. Risk Factors for Recurrence of iMN in Renal Transplantation

In the results of 9 of the 12 studies of the systematic review, the researchers presented various risk factors. Briganti et al. (2002), Ibrahim et al. (2006), and Rodriguez et al. (2012) did not report risk factors, nor was their study designed for this [15,16,20]. No recurrence risk factors were associated using multiple regression analysis by Sprangers et al. (2010) [18]. Female sex, younger age at diagnosis of membranous nephropathy, and high pretransplant proteinuria were presented as risk factors for recurrence by Moroni et al. (2010), Kennedy et al. (2013), and Grupper et al. (2016), respectively [17,21,24]. Finally, five studies indicated that the presence of positive pretransplant anti-PLA2R antibodies is a risk factor for disease recurrence [19,22,23,24,25].

The overall percentage of living donors is 54.44%, and it is confirmed that living-donor transplantation is associated with a statistically significantly increased risk of recurrence of iMN (odds ratio 2.99, *p*-value < 0.001).

### 3.5. Serological Testing and Staining of Biopsies for Anti-PLA2R Antibodies before Transplantation

In this systematic review, 5 of the 12 included studies performed serological screening for anti-PLA2R antibodies before renal transplantation in their population. The aim was to calculate the sensitivity and specificity of this test, as well as the risk of recurrence of primary membranous nephropathy in the presence of anti-PLA2R antibodies. The comparison was made with patients who had a positive serological test of anti-PLA2R antibodies but no recurrence of idiopathic membranous nephropathy. Additionally, in three studies, biopsy staining for anti-PLA2R antibodies was performed pre-transplantation. Specifically, 37 (52.11%) of the 71 patients had a positive serological test for anti-PLA2R antibodies and 18 (25.35%) had a positive biopsy stain. The comparison was made with 11 control patients without disease recurrence and with a positive serological screening. The sensitivity of positive pretransplant serological testing ranges from 57% to 85.30% and the specificity is 85.10–100%, while the positive predictive value and negative predictive value of the test are 83–100% and 42–92%, respectively. Grupper et al. (2016) [24] report that the risk of disease recurrence with a positive pretransplant antibody serological test is greater, with a hazard ratio of 3.761 (95% confidence interval 1.635–8.652).

### 3.6. Outcome of Treatment of Recurrent iMN with Rituximab

A total of 55 (47.82%) of 115 patients (119 grafts) received rituximab as treatment after diagnosis of iMN recurrence. Rituximab was administered in four doses of 375 mg/m^2^ per week, a single dose of 1 g, two doses of 1 g with an interval of 2 weeks, and one to two doses of 375 mg/m^2^ with an interval of two weeks in 8 (14.54%), 2 (3.63%), 19 (34.54%), and 6 (10.90%) patients, respectively. For 20 (36.36%) patients, data on administration are not available. In addition, four patients underwent plasmapheresis at the same time. Treatment outcome was complete and partial remission for 28 (50.90%) and 17 (30.90%) patients, respectively. For nine (16.36%) patients, there was no response to rituximab treatment, while for one (1.81%) patient, no data are available (Table 3).

### 3.7. Graft Survival following iMN Recurrence

In the 134 transplant patients, who received a total of 139 grafts, there was a loss of 30 (21.6%) kidney allografts due to recurrence of iMN. A comparison was made with a control group. The control group consisted of the control groups of the included studies. In the studies in which there is no control group, the transplant patients without recurrence were included as controls. In the total of 138 transplant control patients without iMN recurrence, the loss of 35 (25.4%) grafts was observed due to any other cause. Detailed data on graft loss are presented in Appendix A. The meta-analysis indicated that iMN recurrence was not associated with significantly different risk of graft loss (OR: 1.03, 95% confidence intervals: 0.52 to 2.04) (Figure 2).

No statistical heterogeneity was observed across studies (*I*^2^: 0%); hence, the 95% predictive intervals (0.52 to 2.04) were identical to the confidence intervals. The trim-fill method identified two potentially missing studies in the funnel plot; however, after statistical imputation, the significance of the effect estimate remained unchanged (new OR: 1.25, 95% confidence intervals: 0.66 to 2.35). As a result, no significant risk of small study effects was recognized. Based on the available studies, the power of the meta-analysis was estimated to be 51.8%, 72.9%, and 87.2% for odds ratios of 1.3, 1.4, and 1.5, respectively.

Separate survival comparisons were made between patients with iMN recurrence who were treated with rituximab and those who received no specific treatment based on the possibility that graft survival is not affected by disease recurrence but by response to treatment. There was heterogeneity between the patients of the two groups regarding the interval of occurrence of recurrence and difficulty of examining risk factors among these patients. Recurrence of iMN was not associated with increased risk of graft loss independently of whether patients were treated with rituximab (OR: 0.98, 95% CI: 0.39–2.50, I^2^: 0%) or not (OR: 1.22, 95% CI: 0.58–2.59, I^2^: 3.8%), as shown by the data in Appendix A.

After the diagnosis of recurrence of iMN, the patients who achieved remission (spontaneous, ACEi, immunosuppressive treatment) was compared with those who did not in order to examine whether patients with recurrence of iMN who present remission have better graft survival or not. The achievement of remission is associated with significantly reduced risk of graft loss (OR: 0.14, 95% CI: 0.03 to 0.73) (Figure 3).

## 4. Discussion

### 4.1. Main Outcomes

Whether the impact of recurrence of iMN in a kidney allograft is a significant cause of graft loss remains unclear. Valid diagnosis is required and treatment options should be studied further. The meta-analysis demonstrated a similar risk of graft loss among patients with and without recurrence independently of whether patients were treated with rituximab. However, the achievement of remission is associated with significantly reduced risk of graft loss. The meta-analysis results provide an overall assessment of the association between iMN recurrence and graft survival across all patients included in the analysis. The statement that recurrent disease has no impact on graft survival applies to the overall population studied. Although iMN recurrence itself may not be associated with an increased risk of graft loss overall, achieving remission after treatment appears to be associated with better graft survival. This implies that the impact on graft survival depends on factors such as treatment response and disease management after recurrence. This systematic review highlighted that 21.59% of grafts do not survive due to iMN recurrence. The relative probability of renal allograft loss was calculated and was found to be 1.03 times greater compared to patients without disease recurrence but is not statistically significant. Living-donor transplantation is associated with a statistically significantly increased risk of recurrence of iMN.

The recurrence rate of iMN in the renal graft is estimated to be 30% to 45% [7]. There is difficulty in identifying the true number of patients with disease recurrence because biopsy indications differ between transplant centers and the histological picture of relapse cannot be distinguished from that of de novo membranous nephropathy [26]. The diagnosis of the disease is described to be made between 24 and 36 months post transplantation, but cases of patients have been found in a shorter or longer period [26]. A study in Hong Kong suggests that membranous nephropathy in renal transplantation, including recurrence and the de novo form, is diagnosed at a mean time of 45 months after transplantation [27]. Additionally, iMN recurrence occurs at two time points. The initial clinical manifestation of iMN recurrence may be mild or absent and can only be diagnosed with per-protocol renal biopsies [26]. Its early onset, most likely due to existing circulating anti-PLA2R antibodies, is mainly recognized by protocol biopsies 6 to 12 months after surgery. Late onset, most likely due to new deposition of anti-PLA2R antibodies, is usually recognized 60 months later [7]. This systematic review confirms that in studies with per-protocol biopsies during follow-up, iMN recurrence was diagnosed in fewer months compared to centers that performed clinical-indication biopsies.

Several studies have attempted to identify risk factors for iMN recurrence without success to date [28,29]. The finding of PLA2R antibodies in iMN paved the way for its association with recurrence in renal transplantation. Although finding the specific antibodies pretransplant is considered a risk factor for recurrent iMN, this does not suggest that all seropositive patients will have disease recurrence, nor that all seronegative patients will not show recurrence post transplant. Additionally, other anti-podocyte antibodies, such as THSD7A, may be found in seronegative patients [7]. In a study of the post-transplant period, recurrence of iMN occurred at a mean time of 9.3 months and 18.2 months in recipients of living and deceased donor grafts, respectively, meaning earlier in living donor grafts [30]. Since then, receiving a kidney transplant from a living donor has been reported as a risk factor for recurrence [29], but this is not certain because, in contrast, in the Lyon-Louvain medical school study, disease recurrence was just as common in transplants from deceased donors as from living donors [31], while Dabade et al. observed that, among patients with recurrent iMN, there was a greater number of grafts from deceased donors [32]. Our analysis has identified that receiving a kidney transplant from living donors is a risk factor for recurrence of iMN. Finally, other studies report that risk factors include male gender and rapid progression of initial disease to ESRD [33], data that contradict the conclusions of the systematic review.

Guidelines for the treatment of idiopathic membranous nephropathy are clear for the native kidney but have not yet been established for the treatment of its recurrence in the renal allograft [29]. Symptomatic treatment of iMN recurrence with diuretics, ACEi/ARBs, and hypolipidemic and anticoagulant agents may help manage clinical signs and symptoms, but there is no evidence to confirm that they delay disease progression [34]. In the native kidney, KDIGO guidelines suggest that patients with MN and at least one risk factor for disease progression should be treated with immunosuppression therapy. It is recommended to use rituximab or cyclophosphamide and alternate-month glucocorticoids, or CNI-based therapy, with the choice of treatment depending on the risk estimate. However, transplant patients already are under immunosuppression regimens when the recurrence of iMN occurs. The KDIGO guidelines suggest that transplant patients with recurrent MN should be treated with maximal conservative, antiproteinuric therapy and, for proteinuria >1 g/d, treatment with rituximab is the optimal option. There are no additional data on whether the immunosuppression treatment algorithm for iMN for the general population is a valid option for transplant patients with recurrent iMN, and further studies are needed [35]. Transplant patients are immunocompromised under CNIs, and avoidance of alkylating agents such as cyclophosphamide is deemed appropriate because overimmunosuppression could result in severe infective compilations [8].

The most effective treatment is rituximab, as complete and partial remission of the disease has been observed in 80% [8], an observation that is also confirmed by this systematic review. Several factors need to be considered related to the efficacy of rituximab, such as treatment heterogeneity, sample size, and potential selection bias between patients who received rituximab and those who did not. The use of rituximab should be based on the individual characteristics of each transplant patient. Disease severity, treatment response, comorbidities, and risk–benefit assessment should be considered when determining treatment with rituximab. Further research with larger sample sizes, randomized controlled trials, or observational studies would be needed to better evaluate the efficacy of rituximab in this context. Satisfactory results with the use of rituximab have been described since 2006 in a case with successful treatment of iMN recurrence in which a reduction in proteinuria from 16 g/d to 0.5 g/d was achieved at 3 years [36]. The most common dose of rituximab administration is 375 mg/m^2^ intravenously weekly for four consecutive weeks or, alternatively, two single doses of 1 g intravenously two weeks apart [7], regimens that 27 of the 55 patients in this study received.

The effect of iMN recurrence on renal graft survival has been evaluated by several researchers and remains unclear [7]. The outcome of iMN in the native kidney is not the same as that of its recurrence in the renal graft [26]. Since iMN is considered a progressive disease, with a significant risk of progression even in patients with mild proteinuria, the risk of graft loss is present [7]. Briganti et al. (2002) report that the probability of graft loss due to iMN recurrence at 10 years is 8.4%. Regardless of the effect of disease recurrence, the probability of graft loss over 10 years was similar in patients with a transplant for glomerulopathy compared with those with a transplant for any other reason. In the same study, recurrence of iMN was not associated with an increased risk of graft loss [15]. Similarly, analyzing data from the Renal Allograft Disease Registry, Hariharan et al. (1999) were unable to suggest an increased risk of graft loss in patients with de novo membranous nephropathy and iMN recurrence [37]. Furthermore, this systematic review agrees that glomerulopathy recurrence in the renal graft accounts for 18% to 20% of renal graft loss [38]. However, there is variation in the rate of graft loss specifically attributed to iMN recurrence from 12.5% to 41% at 10 years [39]. Finally, some studies report that 50–65% of renal allografts with iMN recurrence do not survive, while other researchers suggest that there is no statistically significant difference in graft survival comparing patients with iMN recurrence and those without [26,28,33], as is the case in this systematic review.

### 4.2. Strengths and Limitations

The advantages of this systematic review are the examination of data from different studies, the summary of results from the included studies, and the overall larger sample of patients compared to other prospective and retrospective studies. This research does not include case reports or series of patients, which suggests that it is not at risk of selection or publication bias.

There are limitations related to heterogeneity and sample size, follow-up, treatment, design, and type of included studies. Specifically, there is heterogeneity in the sample regarding ethnicity, sample size in each included study, and different comparison patient groups. A limitation related to the follow-up is the different time duration, while the dosage, the duration of rituximab administration, and the combination with other therapeutic measures differ. There is also variation in the definition of the outcome of rituximab treatment because each study gave a different definition of complete and partial remission and no response.

The main result of this systematic review on the relative probability of graft loss is limited to the fact that not all studies were included and concerns 64.58% of the total sample. The difficulty of including the entire sample lies, firstly, in the inability to extract data from studies that do not report graft loss and, secondly, in the inability to calculate the relative probability of loss in studies where there is no loss event. The results related to the role of serologic testing for anti-PLA2R antibodies pre-transplantation are limited by the design of retrospective studies, and sufficient serum samples were not available for all sampled patients. Regarding most of the risk factors for the occurrence of iMN recurrence, these were simply presented in summary from each study included in the systematic review, and no processing or statistical analysis of their data was performed.

## 5. Conclusions

Preliminary data indicate that recurrence of iMN in the kidney graft may not be linked to an increased risk of long-term graft loss given the administration of appropriate treatment, especially rituximab. Patients who do not achieve remission have a higher risk of graft loss. Receiving a kidney transplant from living donors is a risk factor for recurrence of iMN. More prospective studies are needed to establish serological monitoring of anti-PLA2R and other antibodies associated with iMN for disease monitoring and disease outcome post transplant. Rituximab as a therapeutic agent of iMN recurrence seems to show good results; however, the recommended administration dosage and treatment outcome need to be clarified.

## Figures and Tables

**Figure 1 biomedicines-12-00739-f001:**
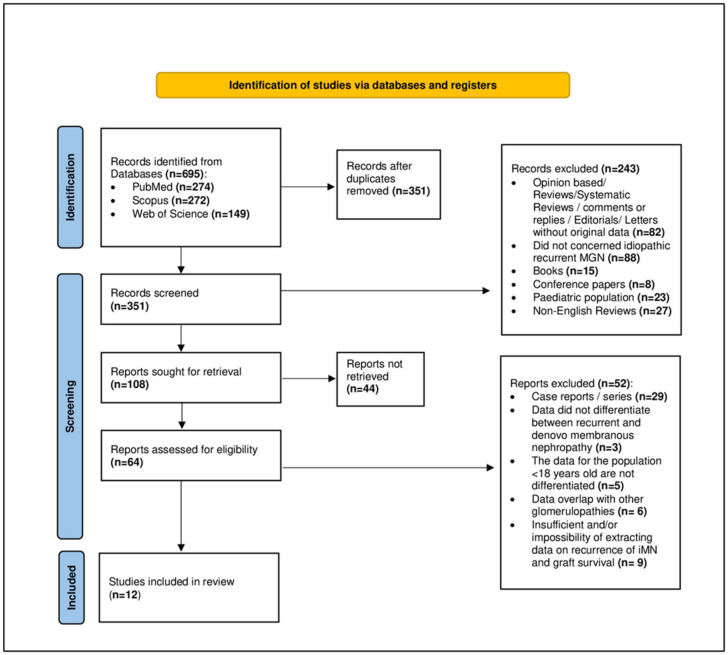
Flow diagram.

**Figure 2 biomedicines-12-00739-f002:**
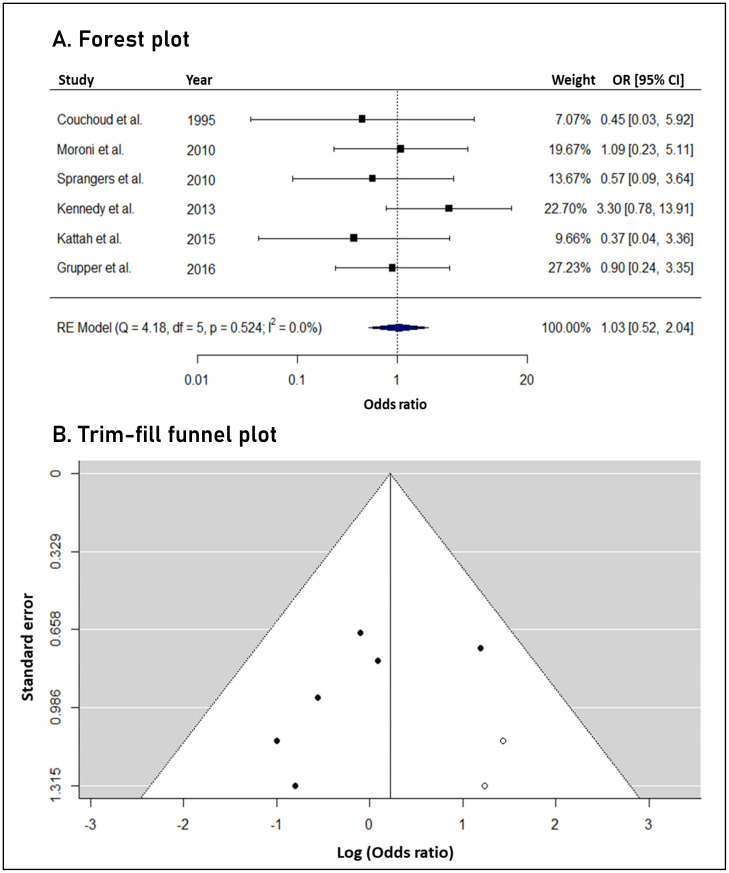
(**A**). Forest plot: iMN recurrence was not associated with significantly different risk of graft loss and no statistical heterogeneity was observed across studies. (**B**). Trim-fill funnel plot: Open cycles indicate possible missing studies, imputed by the trim-fill method [14,17,18,21,22,24].

**Figure 3 biomedicines-12-00739-f003:**
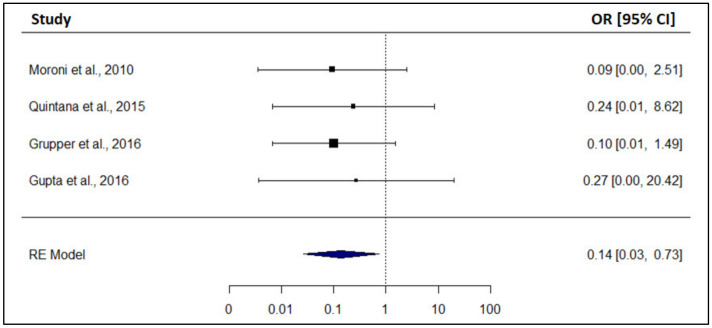
Forest plot: The achievement of remission is associated with significantly reduced risk of graft loss [17,23,24,25].

**Table 1 biomedicines-12-00739-t001:** Overview of studies included in the systematic review.

Study	Study Design	Country	Date of Transplantation	Patients with Diagnosis iMN (*n*)	Patients with Recurrent iMN (*n*)	Grafts with Recurrent iMN (*n*)
Couchoud et al., 1995 [14]	Retrospective	France	January 1980–June 1993	19	5	5
Briganti et al., 2002 [15]	Retrospective	Australia	1988–1997	81	5	5
Ibrahim et al., 2006 [16]	Prospective	USA	10 January 1999–December 2004	19	4	4
Moroni et al., 2010 [17]	Retrospective	Italy	July 1975–July 2009	35	12	12
Sprangers et al., 2010 [18]	Retrospective	USA	1992–2008	34	15	15
Debiec et al., 2011 [19]	Retrospective	France	1982–2006	25	10	11
Rodriguez et al., 2012 [20]	Retrospective	USA	January 2001–February 2011	29	18	18
Kennedy et al., 2013 [21]	Retrospective	Ireland	1982–2010	32	9	13
Kattah et al., 2015 [22]	Retrospective	USA	2000–2010	37	18	18
Quintana et al., 2015 [23]	Retrospective	Spain	January 1994–April 2013	21	7	7
Grupper et al., 2016 [24]	Retrospective	USA	1998–2013	63	30	30
Gupta et al., 2016 [25]	Retrospective	USA	June 2005–January 2014	16	6	6
Total				411	139	144

**Table 2 biomedicines-12-00739-t002:** Median time of iMN recurrence in renal allograft.

Study	Graft with iMN Recurrence (*n*)	Biopsy (*n*)		Follow-Up (Months)	Median Time of iMN Recurrence (Months)
		Per-Protocol	Clinical Indications		
Couchoud et al., 1995 [14]	5	0	5	- ^¥^	39
Ibrahim et al., 2006 [16]	4	0	4	48	18
Moroni et al., 2010 [17]	12	0	12	117	26.2
Sprangers et al., 2010 [18]	15	1	14	70.02	13.6
Debiec et al., 2011 [19]	11	1	9	- ^¥^	15.4
Rodriguez et al., 2012 [20]	18	9	9	35	2.7
Kennedy et al., 2013 [21]	13	0	9	120	46
Kattah et al., 2015 [22]	18	18	0	88	4.1
Quintana et al., 2015 [23]	7	0	7	- ^¥^	22
Grupper et al., 2016 [24]	30	16	14	87.5	3.9
Gupta et al., 2016 [25]	6	0	6	69.6	42
Total	139 *	45	89	35–120	18

^¥^: Not mentioned. *: For 5 patients who received a transplant >1 time, the biopsy method is not mentioned.

**Table 3 biomedicines-12-00739-t003:** Outcome of treatment with rituximab in patients with iMN recurrence.

Study	Patients with iMN Recurrence (*n*)	Patients Treated with Rituximab (*n*)	Outcome		
			Complete Remission	Partial Remission	No Response
Moroni et al., 2010 [17]	12	2	0	1	1
Sprangers et al., 2010 [18]	15	4	3	1	0
Rodriguez et al., 2012 [20]	18	8	8	0	0
Kennedy et al., 2013 [21]	9 *	1	1	0	0
Kattah et al., 2015 [22]	18	11	6	2	3
Quintana et al., 2015 [23] ^¥^	7	6	1	3	1
Grupper et al., 2016 [24]	30	17	9	5	3
Gupta et al., 2016 [25]	6	6	0	5	1
Total	115	55	28 (50.90%)	17 (30.90%)	9 (16.36%)

*: 13 grafts. ^¥^: not available for 1 patient.

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
