# Peer review of "Recurrence of Idiopathic Membranous Nephropathy in the Kidney Allograft: A Systematic Review"

_biomedicines, 2024, doi:10.3390/biomedicines12040739_

Round 1

Reviewer 1 Report

Comments and Suggestions for Authors

The study aims to find current evidence regarding idiopathic membranous nephropathy recurrence in kidney transplantation, focusing on graft survival, risk factors and treatments.

To answer these questions, the Authors performed a systematic review and meta-analysis. The methods are clearly described and also the search criteria, a search on Embase should be performed. A low sample size characterises all the studies retrieved, they were performed over more than 20 years, consequently, their heterogeneity is high and some concerns arise about the opportunity of conducting a meta-analysis. Perhaps a systematic review seems more reasonable.  I suggest also to calculate the power of the meta-analysis.

The results are well presented and the discussion is exhaustive, the limitations are exposed. 

The conclusion is realistic and coherent with the results.

Author Response

Comment:

The study aims to find current evidence regarding idiopathic membranous nephropathy recurrence in kidney transplantation, focusing on graft survival, risk factors and treatments.

To answer these questions, the Authors performed a systematic review and meta-analysis. The methods are clearly described and also the search criteria, a search on Embase should be performed. A low sample size characterises all the studies retrieved, they were performed over more than 20 years, consequently, their heterogeneity is high and some concerns arise about the opportunity of conducting a meta-analysis. Perhaps a systematic review seems more reasonable.  I suggest also to calculate the power of the meta-analysis.

The results are well presented and the discussion is exhaustive, the limitations are exposed. 

The conclusion is realistic and coherent with the results.

Response:

Dear reviewer,

Thank you very much for your comments and suggestions. Below, I hope you find our answers satisfactory and our modifications targeted to your feedback.

Regarding databases, Elsevier, the publisher of both SCOPUS and Embase, integrates Embase content into the SCOPUS database, providing users with access to a broader range of literature within the SCOPUS platform. This integration allows researchers to search and access Embase content alongside other scholarly literature indexed in SCOPUS.

All outcome were initially evaluated qualitatively. Regarding the meta-analysis, the statistical heterogeneity was quantified with inconsistency index, indicating no inter-study heterogeneity (I2: 0%). As a result, the 95% predictive intervals, representing the effects to be expected by future studies, were identical to the 95% confidence intervals. In addition, it has been specified that: “Based on the available studies, the power of the meta-analysis was estimated to be 51.8%, 72.9% and 87.2% for odds ratios of 1.3, 1.4 and 1.5, respectively.” (Page 9).

Please, check the modified manuscript with the corrections in yellow highlight. I wish your suggestions to make our manuscript stronger.

Anastasios Panagakis

Correspondence author

Reviewer 2 Report

Comments and Suggestions for Authors

This is a comprehensive review of a rare, but an important complication seen after transplant of patients with iMN.

Major comments:

Does their meta-analysis support their statement that recurrent disease has no impact on graft survival or is it limited to patients who experience remission after treatment? The fact is that patients who had a complete remission had a better graft survival. 

The other question is related to the efficacy of Rituximab. Does the data support their statement that "graft survival is independent of whether patients were treated with rituximab or not"? There is not enough evidence that the groups with and without Rituximab are comparable, the treatment dose was heterogeneous. Do the authors advocate treatment with Rituximab?

The answers to these two major questions should be included in their conclusion.

Minor comments:

1.     Table 1.  I would add the number of patients with diagnosis of iMN in each series to have a recurrence rate and not just no. of patients with recurrence.

2.     Regarding plasmapheresis and Rituximab, did some of the patients were given treatment prior to transplant to prevent recurrence?

3.       The paragraph on quality assessment (3.3) should be moved up after the study selection as part of the study design before the result section.

4.       One limitation of their analysis of risk factors for recurrence is the small number of patients in some series, therefore I would mention only those factors that are commonly detected in the larger series. So for instance, the series showing a longer interval between diagnosis and ESRD as a risk factor for recurrence is based on only 5 patients with recurrence.

5.       Regarding Rituximab vs. no Rituximab are the patients in the two groups were similar in terms of risk factors for recurrence and interval after transplant for recurrence.

6.       In analysis of graft survival how the control group was selected? Are this group included those with iMN without recurrence reported from all studies. Did all these studies included a control group? Please just make it clear

7.       Page 9 at the bottom "Another analysis was made, to examine whether the patients with recurrence of iMN responded to the treatment." This was mentioned above in section 3.6. Outcome of treatment of recurrent iMN with rituximab.

8.       In table 3. Outcome of treatment with rituximab in patients with iMN recurrence please add percentage in parenthesis.

9.       "The studies included in the systematic review are few and there is heterogeneity." This is mentioned in the same paragraph above (a repeated sentence).

10.   The higher recurrence rate among live donors should be mentioned in the final conclusion paragraph. Does it suggest any genetic cause for the disease?

Author Response

Dear reviewer,

Thank you very much for your comments and suggestions. Below, I hope you find our answers satisfactory and our modifications targeted to your feedback. Please, check the answers below as well as the modified manuscript with the corrections in yellow highlight. I wish your suggestions to make our manuscript stronger.

Anastasios Panagakis

Correspondence author

Major comments:

  1. Comment: Does their meta-analysis support their statement that recurrent disease has no impact on graft survival or is it limited to patients who experience remission after treatment? The fact is that patients who had a complete remission had a better graft survival.

Response: The meta-analysis results provide an overall assessment of the association between iMN recurrence and graft survival across all patients included in the analysis. The statement that recurrent disease has no impact on graft survival apply to the overall population studied. While iMN recurrence itself may not be associated with an increased risk of graft loss overall, achieving remission after treatment appears to be associated with better graft survival. This implies that the impact on graft survival depend on factors such as treatment response and disease management after recurrence.

Please find the suggested modification of this part in Page 10.

  1. Comment: The other question is related to the efficacy of Rituximab. Does the data support their statement that "graft survival is independent of whether patients were treated with rituximab or not"? There is not enough evidence that the groups with and without Rituximab are comparable, the treatment dose was heterogeneous. Do the authors advocate treatment with Rituximab?

Response: Several factors need to be considered related to the efficacy of rituximab such as treatment heterogeneity, sample size and potential selection bias between patients who received rituximab and those who did not. The use of rituximab should be based on the individual characteristics of each transplanted patient. Disease severity, treatment response, comorbidities and risk-benefit assessment should be considered when determining treatment with rituximab.Further research with larger sample sizes, randomized controlled trials or observational studies would be needed to better evaluate the efficacy of rituximab in this context.

Please find the suggested modification of this part in Page 11.

Minor comments:

  1. Comment: Table 1.  I would add the number of patients with diagnosis of iMN in each series to have a recurrence rate and not just no. of patients with recurrence.

Response: The number of patients with iMN is added on table 1. Please find the suggested modification of this part in Pages 4 – 5.

  1. Comment: Regarding plasmapheresis and Rituximab, did some of the patients were given treatment prior to transplant to prevent recurrence?

Response: Some patients were given treatment with plasmapheresis and Rituximab because of ABO incompatibility transplantation, not to prevent iMN recurrence.

  1. Comment: The paragraph on quality assessment (3.3) should be moved up after the study selection as part of the study design before the result section.

Response: Quality assessment is moved in paragraph 3.2. Please find the suggested modification of this part in Page 4.

  1. Comment: One limitation of their analysis of risk factors for recurrence is the small number of patients in some series, therefore I would mention only those factors that are commonly detected in the larger series. So for instance, the series showing a longer interval between diagnosis and ESRD as a risk factor for recurrence is based on only 5 patients with recurrence.

Response: Risk factors mentioned by studies with small number of patients are excluded from the manuscript. Please find the suggested modification of this part in Page 7.

  1. Comment: Regarding Rituximab vs. no Rituximab are the patients in the two groups were similar in terms of risk factors for recurrence and interval after transplant for recurrence.

Response: There is heterogeneity between the patients of the two groups regarding the interval of occurrence of recurrence and difficulty to examine risk factors among these patients. Please find the suggested modification of this part in Page 9.

  1. Comment: In analysis of graft survival how the control group was selected? Are this group included those with iMN without recurrence reported from all studies. Did all these studies included a control group? Please just make it clear

Response: The control group consists of the control groups of the included studies. In the studies that there is no control group, the transplanted patients without recurrence are included as controls. Please find the suggested modification of this part in Page 8.

  1. Comment: Page 9 at the bottom "Another analysis was made, to examine whether the patients with recurrence of iMN responded to the treatment." This was mentioned above in section 3.6. Outcome of treatment of recurrent iMN with rituximab.

Response: This sentence is excluded from the manuscript.

  1. Comment: In table 3. Outcome of treatment with rituximab in patients with iMN recurrence please add percentage in parenthesis.

Response: Please find the suggested modification of this part in Page 8.

  1. Comment: "The studies included in the systematic review are few and there is heterogeneity." This is mentioned in the same paragraph above (a repeated sentence).

Response: This sentence is excluded from the manuscript.

  1. Comment: The higher recurrence rate among live donors should be mentioned in the final conclusion paragraph. Does it suggest any genetic cause for the disease?

Response: The fact that receiving a kidney transplant from living donors is a risk factor for recurrence of iMN does not directly suggest a genetic cause for the disease. While genetics may play a role in predisposing individuals to autoimmune diseases like iMN, other factors such as immune system dysregulation, environmental triggers, and immunologic compatibility between the donor and recipient could also contribute to the recurrence of iMN post-transplantation. Please find the suggested modification of this part in Page 12.